# Echo chambers can emerge without algorithmic personalization or a preference for homogeneity

Petter Törnberg *

ILLC, University of Amsterdam, Amsterdam, The Netherlands

* p.tornberg@uva.nl

## Abstract

Online ideological segregation—often described as "echo chambers"—is commonly attributed to algorithmic personalization ("filter bubbles") or users' preferences for like-minded environments. We propose a different mechanism. Using a minimal agent-based model, we show that strong segregation can arise even without algorithmic personalization and without users preferring homogeneous environments. Even when users exit communities only after finding themselves almost entirely surrounded by disagreement, cascading exits can push initially mixed communities toward high homogeneity. Once small imbalances arise, feedback between exit and regrouping generates a self-reinforcing process of system-level sorting. Extending the model further reveals that algorithmic personalization can, under some conditions, *reduce* segregation by lowering dissatisfaction, slowing exit cascades, and stabilizing mixed communities. As an empirical illustration, a longitudinal analysis of the subreddit `r/MensRights` shows that users whose language is more distant from the community's evolving semantic center are more likely to exit. Taken together, these findings suggest that echo chambers need not depend on users seeking homogeneous environments or algorithmic personalization alone, but can also emerge from exit dynamics in the interaction structures characteristic of online platforms. More broadly, they show how interventions aimed at individual exposure can produce aggregate dynamics that differ from their intended effects, complicating both scientific and policy debates over online polarization.

## 1 Introduction

Ideologically homogeneous online environments—variously termed echo chambers [1], deliberative enclaves [2], cyberbalkanization [3], or filter bubbles [4]—are widely viewed as among the most consequential transformations of the digital public sphere. They have been linked to rising polarization [5], radicalization [6], stronger confirmation biases [7,8], the spread of misinformation [9], and deepening divergence in citizens' fundamental worldviews [10]. Although scholars debate the extent and prevalence of online homophily [11–13] and its downstream consequences [6,14–16],

**Data availability statement:** http://github.com/cssmodels/onlinesegregation.

**Funding:** Dutch Research Council (NWO) VIDI Grant VI.Vidi.231S.089.

**Competing interests:** The authors have declared that no competing interests exist.

there is broad agreement that many digital platforms exhibit strong tendencies toward ideological clustering [17]. Emerging research further shows that fragmentation increasingly occurs *across* platforms: entire platforms now attract ideologically distinct audiences, producing system-level "echo platforms" rather than merely echo chambers within them [18,19].

What causes these patterns of ideological sorting remains a central—and unresolved—question. Two dominant explanations prevail. The first emphasizes *algorithmic curation*: recommender systems and ranking algorithms on platforms such as Facebook, TikTok, and YouTube structure users' informational environments in ways that can narrow ideological exposure, often without users' awareness [4,13,20]. From this perspective, ideological homogeneity is a design-driven phenomenon.

The second highlights *selective exposure*, arguing that users themselves choose homogeneous spaces, curating their environments by seeking agreeable content and avoiding dissonant viewpoints [7,21]. Here, homogeneity emerges from user-driven behavioral tendencies rooted in confirmation bias and social identity processes [5,10]. Empirical evidence often lends weight to this position: studies suggest that user-level homophily often exceeds algorithmic effects [22], partisan engagement persists even when algorithmic amplification is constrained [23], and disabling algorithmic feeds yields only modest changes in polarization [24]. These findings suggest that individual choice, rather than algorithmic intervention, may be the primary engine of online ideological sorting.

This paper offers a different explanation. We show that ideological homogeneity can arise *even when users tolerate substantial disagreement and no algorithmic personalization is present*. Using a minimal agent-based model, we demonstrate that integrated communities are dynamically unstable: simple dissatisfaction-and-exit dynamics generate endogenous feedback loops that push systems toward homogeneity. Even weak thresholds are sufficient to trigger cascading realignments across communities—without strong homophilic preferences, algorithmic personalization, or deliberate self-sorting. The interaction topologies of online communities can themselves generate ideological homogeneity, even in the absence of algorithmic personalization or preference for homogeneous environments. We further uncover an effect that complicates prevailing assumptions. Although algorithmic filtering is widely viewed as a driver of echo chambers, exposing users to more like-minded content can, in this structural context, *reduce* segregation by stabilizing otherwise fragile mixed communities [25].

To assess whether this micro-level mechanism also appears in real online communities, we conduct an illustrative analysis using data from the large and long-running subreddit `r/MensRights`. The community originated as a relatively heterogeneous space but gradually evolved into one of the central echo chambers of the "manosphere" [26]. Leveraging complete comment histories from 2007–2015, we construct a user–month panel and estimate whether users whose language diverges from the subreddit's center of gravity are more likely to exit. The results are compatible with the model's core assumption: users who are more distant from the community's evolving center are more likely to exit. While descriptive rather than causal, this

pattern provides suggestive evidence that the dissatisfaction-exit mechanism formalized in the model operates in real group-based online environments.

Taken together, these findings broaden how we theorize ideological homogeneity online. In addition to accounts centered on intentional design and user choice, they point to segregation as an emergent outcome of how social interaction is organized in online environments. This structural perspective clarifies why echo chambers persist even when algorithms are disabled or users seek diverse content, and why interventions that ignore systemic feedback dynamics may fail—or even backfire. The challenges of social media may lie less in specific algorithmic decisions than in the very architectures through which digital sociality is organized.

This study makes three contributions. First, we introduce a minimal agent-based model of group-based online interaction that extends Schelling's segregation dynamics to contemporary platform architectures, showing that integrated communities are dynamically unstable even when users tolerate substantial disagreement. Second, we show that algorithmic personalization can, under plausible conditions, reduce segregation by dampening exit dynamics, challenging dominant narratives about "filter bubbles". Third, we provide an illustrative empirical plausibility check using longitudinal data from r/MensRights, showing that users who are linguistically distant from a community's evolving center are more likely to exit — a pattern consistent with the model's core micro-level assumption.

## 2 A dynamic model of online segregation

A large literature on online segregation and echo chambers has documented how digital publics fragment into ideologically aligned clusters, but it offers divergent accounts of the mechanisms behind this process. Empirical studies show that users' information environments on platforms such as Facebook, Twitter/X, and YouTube are often skewed toward like-minded content, yet the strength and pervasiveness of such "echo chambers" remain contested [11–17]. Two broad explanations dominate: one emphasizes algorithmic curation and "filter bubbles," arguing that recommender systems systematically narrow exposure to cross-cutting views [4,20,27,28]; the other stresses selective exposure and homophily, highlighting users' tendency to seek attitude-consistent content and social ties [1,7,10,21]. Formal models build on these intuitions, typically generating segregation through homophilic rewiring in adaptive networks or through algorithms that preferentially surface congruent content [25,29–34]. Across these approaches, homogeneity is usually treated as the outcome of either intentional sorting by users or deliberate amplification by platforms. In what follows, we take a different route: we ask whether strong ideological clustering can arise even when users tolerate substantial disagreement and algorithms play no role, but rather as a consequence of the group-based interaction architectures that structure online life.

Against this backdrop, Schelling's [35] seminal model of residential segregation offers a classic demonstration of how weak micro-level preferences can generate strong, unintended macro-level separation. In his model, agents of two types (e.g., red and blue, often interpreted as racial groups) occupy positions on a toroidal lattice representing an urban area. At each time step, agents evaluate the composition of their local Moore neighborhood; if the proportion of same-type neighbors falls below a specified tolerance threshold, they relocate to a randomly selected vacant site. This process repeats until all agents are satisfied or the system reaches a dynamic equilibrium. Despite its simplicity, the model yields a striking result: even when agents are highly tolerant of diversity—requiring, for instance, only 30% of neighbors to be like themselves—the system nonetheless evolves toward near-complete segregation. We refer to this emergent pattern, in which observed homophily substantially exceeds individual preferences, as the *Schelling segregation effect*.

The mechanism underlying this effect is a self-reinforcing feedback loop. As agents relocate, they alter the composition of their neighborhoods, triggering dissatisfaction and further movement by others. Mixed configurations are theoretically stable but dynamically fragile: once small local imbalances emerge, they cascade toward segregated outcomes. This central insight—that strong large-scale outcomes can emerge from weak micro-level causes—has been widely applied across disciplines, from urban sociology to opinion dynamics and even phase separation in physics.

However, Schelling's original model is grounded in a spatial metaphor: agents interact with physical neighbors in geographic proximity, and relocation means moving to a new place in that spatial landscape. Although this captures many offline phenomena, it does not reflect the dominant interaction structures of online environments. Digital platforms organize interaction through networks and bounded groups or communities. While these communities may be influenced by offline spatial relations [36], they operate under different interaction topologies: exiting a group or rewiring network ties is not equivalent to moving on a spatial lattice. Because the topology of interaction fundamentally shapes the social dynamics that unfold, this shift—from geographic neighborhoods to modular, bounded communities—represents a major transformation introduced by digital technologies. A key question, therefore, is whether the core mechanism identified by Schelling persists within these qualitatively different online architectures.

A growing body of research has extended the insight that segregation can emerge from simple local preferences into the study of online opinion dynamics. Rather than modeling spatial relocation, these studies focus on networked settings in which ties form, dissolve, and rewire based on similarity. Classic coevolutionary models show that even weak homophily in tie formation can lead to spontaneous clustering, as agents disconnect from dissimilar others and preferentially connect to like-minded peers [29–31,37]. Others highlight how biased assimilation and selective exposure amplify these processes, further reducing cross-cutting ties even when preferences remain moderate [32,38,39]. Another strand of research examines how algorithmic curation interacts with homophilic behavior, accelerating segregation by reinforcing existing preferences and filtering out dissenting content [27,28].

While this literature has yielded important insights, it generally assumes either explicit homophily or algorithmic filtering as necessary preconditions for segregation. In these network models, homogeneity arises not through Schelling-like cascades but through iterated homophilic rewiring: agents sever ties to dissimilar others and form new ones with similar peers, leading to increasingly homogeneous clusters over time [29–31,37]. The classic Schelling feedback dynamic — in which each relocation alters local composition and triggers further moves — is typically absent. As a result, these models offer a fundamentally *intent-driven* view of segregation: homogeneity results from deliberate user choices or engineered recommender systems.

Here, we shift the focus from spatial and network-based dynamics to *group-based interaction structures* — a topology that better captures the architecture of many online platforms, such as forums, group chats, and communities, as well as the dynamic across platforms, yet remains underexplored in segregation modeling. Our goal is to examine whether a classic Schelling segregation effect — defined as a cascading feedback loop triggered by local dissatisfaction — can occur in these environments even without algorithmic filtering or strong homophily preferences.

In our model, $N$ agents are distributed uniformly at random across $C$ discrete communities, representing bounded digital spaces such as online forums or chat groups. Each agent holds a fixed binary opinion, $o \in \{0, 1\}$, assigned at random.

At each time step, for every agent, the model draws a focal agent at random and has them interact with $k$ randomly sampled members of their current community. The focal agent then computes the share $f$ of interactors who hold the same opinion. If $f \leq \theta$, where $\theta$ represents the agent's tolerance for disagreement, the agent becomes dissatisfied and relocates to a randomly selected new community. Similarly to the Schelling model, we deliberately assume that relocation is *uninformed*: agents do not observe the composition of other communities before moving and have no intrinsic preference for any particular community. This simplifies the choice environment and allows us to isolate a minimal mechanism: whether segregation can arise even when agents are not actively searching for like-minded spaces, but merely exiting groups in which they feel too isolated. (In real settings, where users often do gravitate toward communities that appear more congenial, the segregation dynamics would likely be even stronger.) The model therefore isolates a minimal mechanism through which ideological sorting can emerge in group-based online environments. Agents do not change their opinion and have no intrinsic preference for any particular group—relocation is governed solely by minimal dissatisfaction with local composition.

 

Although highly stylized, each component of the model captures key empirical features of online behavior. Users typically interact with only a subset of a community, not its full membership. They frequently exit groups, forums, or chat spaces when they feel unwelcome, isolated, or outnumbered, often migrating to alternative communities more aligned with their views. The tolerance threshold reflects well-documented patterns of selective participation, where users exit once disagreement exceeds a certain level, even if they do not seek perfect ideological conformity. The assumption of binary opinions, standard in the modeling literature, isolates the effect of group composition on sorting dynamics. Together, these assumptions allow us to focus on a central question: can micro-level dissatisfaction alone, in the absence of algorithmic manipulation or strong preferences, generate large-scale ideological segregation?

To quantify ideological sorting, we define the segregation index $S \in [0, 1]$ as the expected probability that a random within-community interaction occurs between agents holding the same opinion. Formally,

$$S = \mathbb{E}\left[\Pr\left(o_i = o_j \mid i, j \in c\right)\right].$$

A value of $S = 1$ indicates complete ideological sorting (no cross-opinion encounters), whereas $S = 0$ corresponds to fully mixed communities (opinions are random with respect to community). We moreover define $\phi$ as the final segregation level at the end of the simulation horizon.

This minimal model allows us to investigate whether online homogeneity can emerge purely from the architecture and dynamics of bounded group interaction — even in the absence of both algorithmic curation and strong user preferences for like-mindedness.

We now turn to the results, examining the conditions under which segregation emerges and how these patterns compare to classical Schelling dynamics.

## 3 Results

Throughout this section, unless otherwise stated, we use a baseline configuration with $N = 2,000$, $C = 20$, and interaction sample size $k = 10$. For each parameter combination we run the model for 50 time steps (i.e., $10^5$ agent updates).

Fig 1 shows the temporal evolution of average community opinion across varying values of the tolerance parameter $\theta$. When $\theta = 0$ or is very low, agents are almost never dissatisfied with their group composition, and no meaningful segregation emerges. As $\theta$ increases, however, a clear transition becomes evident. Around $\theta = 0.12$, the system begins to exhibit strong polarization, and by $\theta = 0.2$ complete segregation occurs rapidly, often within just a few iterations. This pattern illustrates a central result: even modest increases in intolerance to dissimilar opinions can trigger rapid system-level shifts from integrated to polarized states.

Fig 2 further highlights this nonlinearity by plotting the final segregation level $\phi$ as a function of $\theta$. The results reveal a tipping point around $\theta \approx 0.1$, beyond which the system reliably converges to highly segregated states. Importantly, this threshold is substantially lower than in the classical spatial Schelling model, where segregation typically emerges only when tolerance levels exceed ~0.3 [35]. This suggests that community-based interaction structures characteristic of online platforms amplify segregation dynamics, allowing large-scale homogeneity to emerge under much weaker conditions.

To assess the robustness of the model's dynamics, we conducted a sensitivity analysis varying both the total number of agents $N$ and the number of communities $C$. As shown in Fig 3, the segregation trajectories are highly stable across a wide range of population sizes and community counts. Increasing $N$ or $C$ does not alter the qualitative dynamics of the model: segregation rises steadily over time and converges toward similar levels in all cases.

To understand why such strong segregation arises under minimal dissatisfaction, we analyze the migration probability analytically. Consider an agent placed in a fully integrated community, where opinions are equally distributed. The agent samples $k$ peers and will migrate if the fraction of similar opinions $H$ is less than or equal to $\theta$. This situation can be modeled as a binomial probability:

$$\Pr\left(\text{migrate}\right) = \Pr\left(H \leq \lfloor \theta \cdot k \rfloor\right) = \sum_{i=0}^{\lfloor \theta \cdot k \rfloor} \binom{k}{i}(0.5)^i(0.5)^{k-i} = \sum_{i=0}^{\lfloor \theta \cdot k \rfloor} \binom{k}{i}(0.5)^k$$

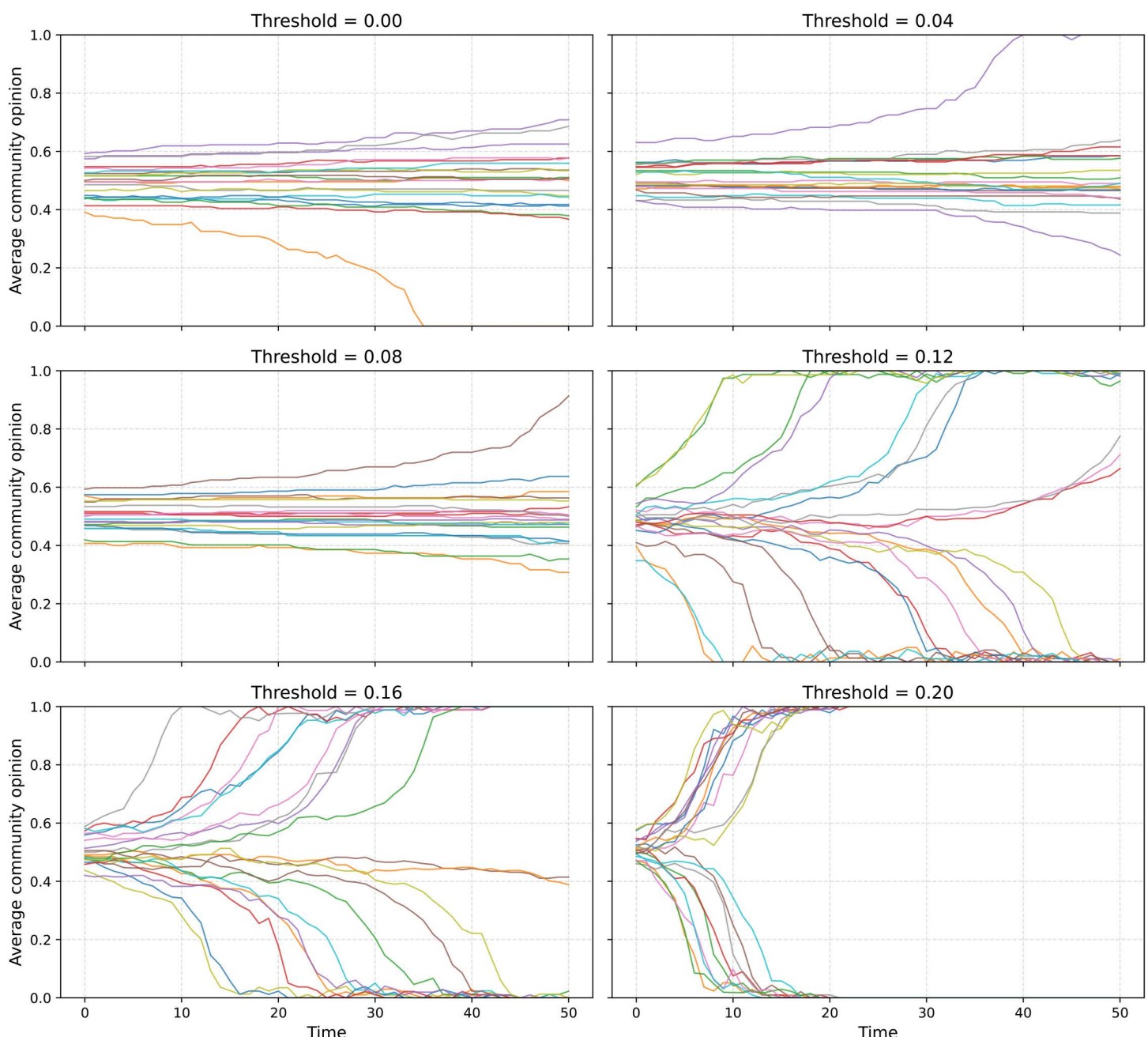

**Fig 1. The evolution of the average opinions of each community as a function of $\theta$, i.e., the threshold by which the agents move if fewer than this number of their interactors are of the same type.** Parameters are $N=2,000$, $C=20$, $k=10$. We run the model 50 steps; while the model does not necessarily reach a stable equilibrium during this time, this allows capturing the strength of the segregation mechanisms—and whether it is likely to be realized. As can be seen, already when the agents require only 12% of the interactors to be of their own type, nearly complete macro-level segregation results.

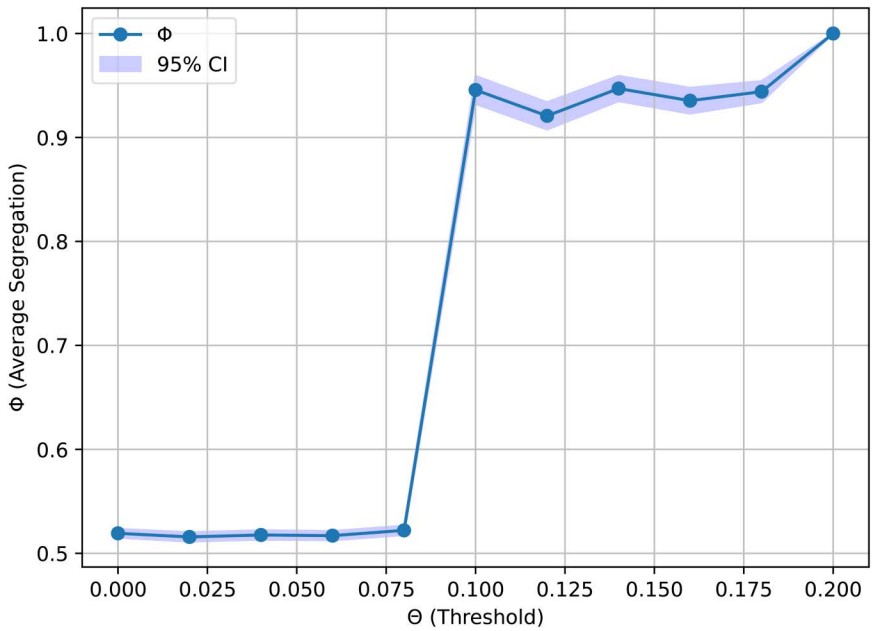

**Fig 2. The final segregation value as a function of $\theta$, using the same parameters as Fig 1.** The model was run 30 times for each data point, and the results are averaged over each run, and shown with 0.95 confidence intervals. The figure suggests that even when users require merely 10% of their interactors to be of the same type as themselves, nearly complete segregation results.

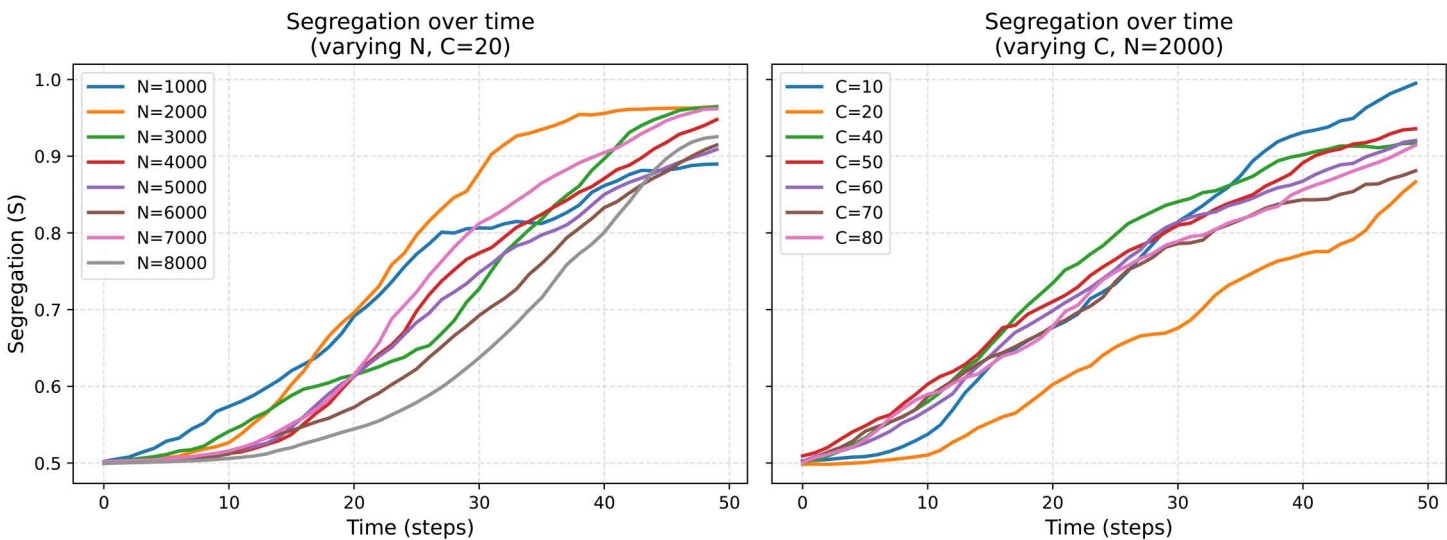

**Fig 3. Sensitivity of segregation dynamics to population size and number of communities.** The plots show the evolution of segregation $S$ over time under different model sizes. Left: Total population $N$ varies from 1,000 to 8,000 while the number of communities is fixed at 20. Right: The number of communities varies from $C=10$ to $C=80$, while the total population is fixed at $N=2,000$. In all conditions, segregation increases monotonically and follows a highly similar trajectory. Differences in the early dynamics reflect stochastic variation rather than systematic effects of system size. The results suggest that the qualitative behavior of the model is invariant to changes in total population and number of communities.

This calculation reveals the mechanism driving segregation: because agents evaluate their environment based on a limited sample of interactions, even small random fluctuations can cause the observed share of similar peers to fall below the threshold, triggering relocation. Once exits begin, they subtly change the composition of both the origin and destination communities, further increasing dissatisfaction for some and satisfaction for others. This self-reinforcing process transforms minor stochastic variation into large-scale ideological sorting. Crucially, it occurs under conditions far weaker than those required in spatial models, suggesting that online community architectures are inherently predisposed to tipping dynamics.

Fig 4 illustrates this analytically derived probability landscape. A discrete jump in migration probability appears around $\theta = 0.1$ for $k = 10$. When $\theta < 0.1$, migration is extremely rare: the agent will only move if they observe no similar opinions, yielding a probability of 0.00098. Once $\theta > 0.1$, however, observing even one similar peer becomes insufficient, and the migration probability jumps to 0.01074 — an order-of-magnitude increase. This discontinuity marks the onset of a cascading process: agents in slight minorities are more likely to exit, incrementally reshaping group compositions and amplifying local majorities. Over time, this feedback loop produces near-complete ideological sorting, even when initial conditions are fully integrated and individuals tolerate substantial disagreement.

### 3.1 Filter bubbles and segregation dynamics

To examine how algorithmic curation shapes community dynamics, we extend the baseline model with a simple *filter-bubble mechanism* that modifies what agents *see* without altering the underlying composition of communities. We introduce a parameter $b$ that governs the share of interaction partners who are artificially selected to hold the same opinion as the focal agent. In each interaction, a fraction $b$ of the $k$ sampled partners is drawn uniformly from same-opinion agents within the community, while the remaining $1 - b$

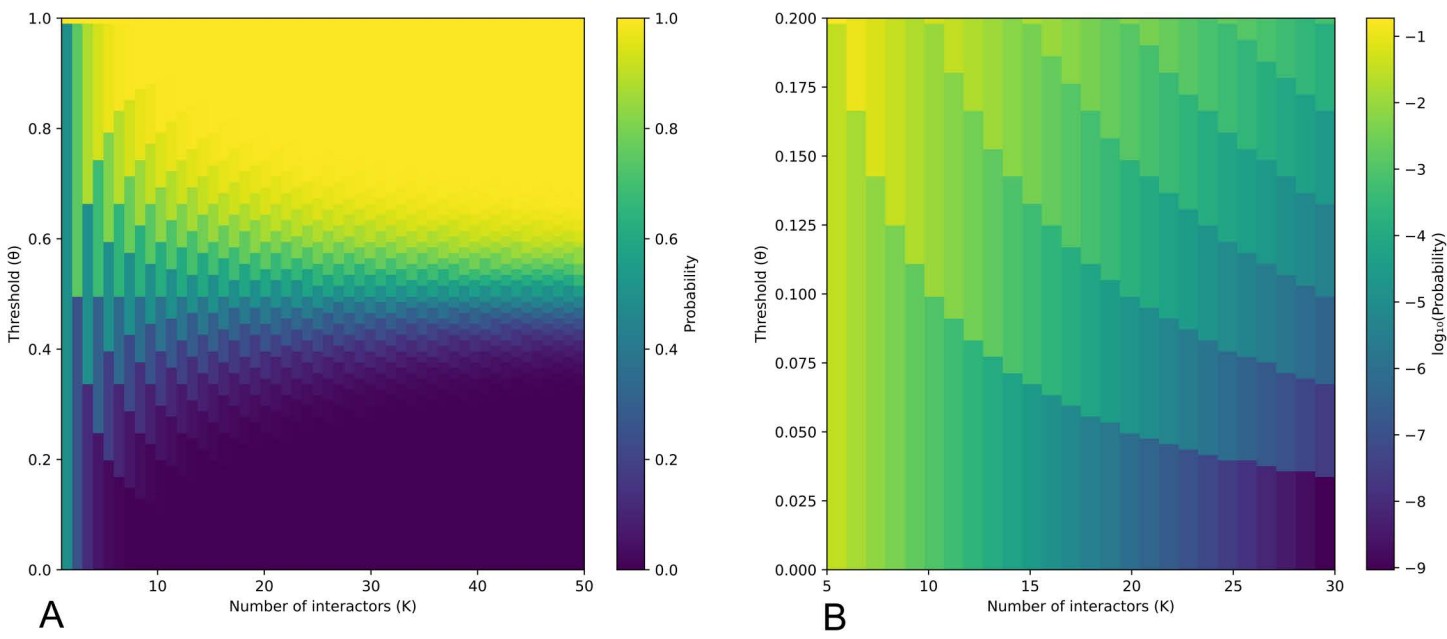

**Fig 4. The analytical solution to the probability of an agent moving from a fully integrated community as a function $k$ and $\theta$.** Figure A shows the full landscape. Figure B shows the log of the resulting probability, zooming in on the relevant part of the landscape. Given 50 step runs, a probability of around $10^{-2}$ is sufficient to enable a tipping point and drive complete segregation. Given 50 step runs, a probability of around $10^{-2}$ is sufficient to enable a tipping point and drive complete segregation.

are sampled randomly, as in the baseline model. This changes only the distribution of opinions in a user's *experienced* interaction set, not the actual opinions or membership of the community.

This abstraction captures the role of algorithmic ranking on contemporary platforms. Even in community-driven environments such as Reddit, personalization affects which posts a user is shown—both on the home feed and within subreddits—based on past engagement. Because users typically see only a small subset of available posts, such ranking systems can increase the likelihood of encountering like-minded content even in otherwise diverse communities. The parameter $b$ therefore represents curated *exposure*, reflecting widely discussed mechanisms of selective amplification on digital platforms.

Such "filter bubbles" are widely understood to intensify online homophily by increasing exposure to like-minded content and reducing opportunities for cross-cutting interaction [4]. Prior modeling work has examined how algorithmic filtering interacts with segregation dynamics, with results that depend on the underlying mechanism. In some settings, filtering reinforces echo chambers by amplifying existing similarity patterns [33,34]. In others, filtering can slow segregation by reducing disagreeing encounters; for example, Pansanella et al. [25] show this in a network-rewiring model.

The addition of algorithmic filtering requires us to analytically separate two forms of segregation. We define *perceived final segregation* ($\phi_{\text{perc}}$) as the expected probability that a random interaction in the curated environment occurs between agents holding the same opinion, thereby capturing the user's experienced interaction environment. In contrast, *structural segregation* ($\phi_{\text{struct}}$) measures the actual degree of ideological sorting in the underlying community, independent of algorithmic intervention.

The results, shown in Fig 5, reveal a notable dynamic: stronger algorithmic filtering reduces agents' exposure to disagreeing opinions and therefore lowers their propensity to relocate. The result of this is that *both* structural and perceived segregation decline sharply as filtering strength $b$ increases. Although filtering slightly raises the share of like-minded

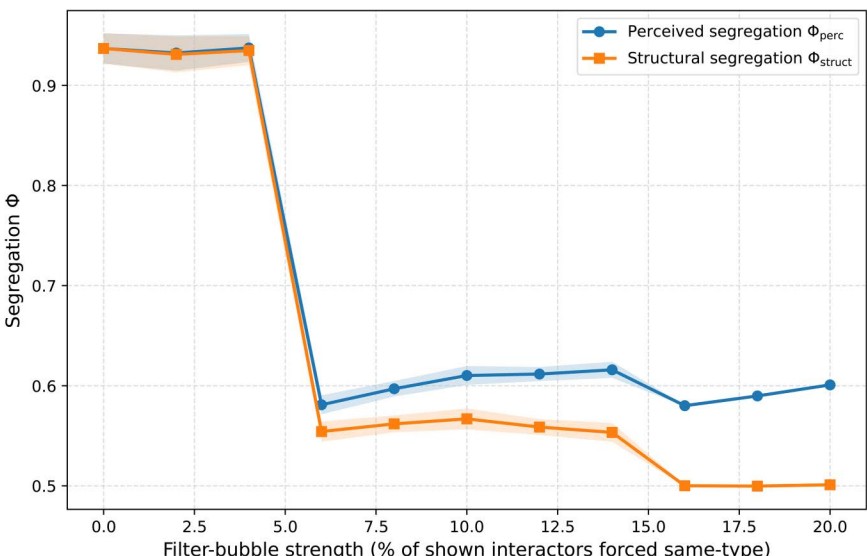

**Fig 5. Effects of algorithmic filtering on perceived and structural segregation.** The model was run 30 times for each parameter value, and segregation is shown averaged over the final segregation value, and shown with 0.95 confidence interval. Parameters are $N = 2,000$, $C = 20$, $k = 10$, $\theta = 0.1$. We run the model 50 steps. The figure shows the mean final segregation $\phi$ as a function of filter-bubble strength $b$, defined as the percentage of within-community interactors that are artificially selected to hold the same opinion as the focal agent. The blue line shows *perceived final segregation* $\phi_{\text{perc}}$, which captures the probability that users encounter same-opinion content within their curated interaction environment. The orange line shows *structural final segregation* $\phi_{\text{struct}}$, which captures the actual degree of ideological sorting in the underlying community structure. Both measures decrease sharply as $b$ increases, indicating that algorithmic filtering reduces not only macro-level sorting but also users' experienced homogeneity.

content within a given community, this local effect is overwhelmingly outweighed by the dampening of exit-driven sorting. By making agents more satisfied with their immediate informational environment, filtering suppresses the cascading exit-and-realignment dynamics that would otherwise push communities toward homogeneity. As a result, communities become more ideologically mixed, and users experience *more*, rather than less, cross-cutting interaction.

This finding carries significant theoretical implications. Algorithmic curation that appears to narrow informational diversity may, under certain conditions, *increase* the diversity of users' actual interaction environments by disrupting the feedback loops that generate segregation in the first place.

Efforts to algorithmically expose users to difference can sometimes backfire: by heightening dissatisfaction, they may intensify the very exit-and-realignment dynamics that drive segregation. By contrast, modest levels of "filter bubbles" can stabilize mixed communities and thereby foster greater heterogeneity. More broadly, this result highlights a key principle: emergent outcomes in digital systems are shaped less by individual intentions or platform design choices than by the structural feedback dynamics through which micro-level behaviors aggregate. These dynamics can invert intuitive expectations, causing well-intentioned interventions to produce precisely the opposite of their desired effects.

### 3.2 Empirical illustration of exit dynamics on `r/MensRights`

A core assumption of the model is that users who are poorly aligned with a community are more likely to exit. To assess the empirical plausibility of this mechanism, we provide an illustrative analysis of the Reddit community `r/MensRights`, testing whether linguistic distance from the community's monthly centroid is associated with subsequent exit.

`r/MensRights` is a large, long-running subreddit that has played a prominent role in what is often described as the "manosphere" [26]. Prior research documents a marked ideological shift during the early 2010s, as the community increasingly adopted oppositional framings of gender politics and became more internally homogeneous. The subreddit is therefore frequently cited as an example of an online echo chamber, making it a useful setting for illustrating how ideological homogeneity can emerge over time.

Several features make `r/MensRights` particularly informative for this purpose. First, the subreddit initially attracted a relatively heterogeneous user base with diverse perspectives on gender-related issues. This early pluralism allows us to observe whether exit patterns consistent with a Schelling-like segregation dynamic emerge as the community evolves. Second, the community is sufficiently large and active to permit fine-grained temporal analysis. The availability of complete comment histories allows us to reconstruct monthly participation trajectories for thousands of users across multiple years, enabling us to examine whether individuals whose language increasingly diverges from the community's evolving center of gravity are more likely to exit.

To evaluate this mechanism, we collect all comments in the community from PushShift, and construct a user–month panel covering all activity in `r/MensRights` from its founding in 2008–2015—a period in which the subreddit experienced substantial ideological consolidation. This dataset allows us to assess the core micro-level assumption of our model: users who are linguistically distant from the community's center are more likely to exit.

We parse the complete Reddit comment archive for `r/MensRights`. For each comment, we extract the author, timestamp, and comment text. We then aggregate all comments by user and month, concatenating a user's comments within each month into a single document. To ensure stable linguistic representations and to focus on genuine participants, we restrict the analysis to users who produced at least five comments in total and who were active in at least two distinct months. This filtering step removes one-off or transitory contributors that would otherwise inflate exit rates and introduce noise.

For each user–month document, we compute a vector representation using a sentence-transformer model (`all-MiniLM-L6-v2`). Embeddings are L2-normalized. Let $\mathbf{v}_{u,t}$ denote the embedding for user $u$ in month $t$. For each month $t$, we compute a community centroid

$$\mathbf{c}_t \;=\; \frac{1}{N_t}\sum_{u \in A_t} \mathbf{v}_{u,t},$$

where $A_t$ is the set of active users in month $t$.

A user's semantic distance from the community in month $t$ is defined as the cosine distance between their embedding and the monthly centroid:

$$d_{u,t} \;=\; 1 - \langle \mathbf{v}_{u,t},\, \mathbf{c}_t \rangle.$$

Because embeddings are normalized, this distance measure corresponds directly to linguistic or semantic divergence from the community's center of gravity.

We define a user as *active* in month $t$ if they post at least once in that month. A user is coded as *exiting* in month $t$ if $t$ is their final observed month of activity and they make no further posts in the subsequent six months. For each user–month observation, we therefore assign a binary outcome variable $\text{exit}_{u,t+1:t+6} \in \{0, 1\}$.

We estimate a logistic regression in which subsequent exit is regressed on users' linguistic distance from the community:

$$\Pr(\text{exit}_{u,t+1:t+6} = 1) \;=\; \text{logit}^{-1}\!\left( \alpha + \beta\, z(d_{u,t}) + \gamma\, z(\text{tenure}_{u,t}) + \delta_{\text{year}(t)} \right).$$

All continuous predictors are z-standardized (mean 0, standard deviation 1), so that each coefficient can be interpreted as the change in the log-odds of exit associated with a one-standard-deviation increase in the corresponding variable. We include year fixed effects $\delta_{\text{year}(t)}$ to control for temporal changes in overall exit rates. Tenure is defined as the number of months since a user's first appearance in the subreddit, allowing us to separate semantic distance from natural turnover dynamics.

**3.2.1 Results.** Linguistic distance from the monthly community centroid is positively associated with exit. The coefficient on standardized linguistic distance is positive and statistically significant ($\beta \approx 0.153$, $p < 0.001$), corresponding to an odds ratio of approximately 1.17. Substantively, a one–standard-deviation increase in distance from the monthly community centroid is associated with a 16–18% increase in the odds of exiting within the subsequent six months, net of tenure and year fixed effects. This pattern is also visible in nonparametric form. Exit probabilities increase monotonically across quartiles of the distance distribution, rising from 13.5% among users most closely aligned with the community's linguistic center to 20.0% among those furthest from it.

Several limitations should be noted. First, the regression does not include user fixed effects or random effects, so the estimated association may partly reflect stable differences in engagement style or propensity to exit rather than dissatisfaction with community composition. Second, linguistic distance from the centroid conflates ideological disagreement with other sources of divergence, such as differences in topic choice or rhetorical style, and is itself shaped endogenously by the participation and exit patterns it is meant to predict. Third, exit is behaviorally ambiguous: cessation of posting may reflect reduced platform use, migration to other communities, or factors entirely unrelated to ideological misalignment. Fourth, the observations are not fully independent, as the same users contribute repeated observations over time, exit is an absorbing state, and distances are jointly determined through a shared monthly centroid; these features violate the conditional independence assumption of standard logistic regression and may affect the precision of the estimated coefficients.

The results are compatible with the model's core assumption: individuals who are more linguistically distant from the community's evolving center are more likely to exit. This dynamic is consistent with a Schelling-like segregation mechanism in which modest individual-level misalignment, amplified through exit rather than overt preference for homogeneity,

produces increasing ideological consolidation at the community level over time. While descriptive rather than causal, this pattern provides an illustrative plausibility check for the model's micro-level assumption that users whose language diverges from a community's center are more likely to exit.

## 4 Discussion and conclusion

A longstanding debate in the literature attributes online ideological segregation either to algorithmic filtering or to users' preferences for homogeneous environments. This study demonstrates that neither is necessary. We show that strong ideological segregation can emerge endogenously from group-based interaction structures themselves. Even weak dissatisfaction with local misalignment can trigger tipping dynamics that drive communities away from integrated states and toward internally homogeneous configurations. Once a slight majority forms within a group, minority members become more likely to exit, further reinforcing homogeneity and potentially seeding similarly aligned groups elsewhere. The cumulative result is a bifurcation of the system into segmented and mutually alienated communities, arising without centralized coordination, strong ideological motivation, or algorithmic manipulation.

This mechanism differs in important ways from most prior extensions of Schelling's insight to online environments, which have primarily focused on networked interaction structures such as follower graphs or friendship ties [30–32]. In networks, tie formation and dissolution affect only local connections and typically generate gradual segregation through homophily—agents selectively connect to similar others or cut ties with dissimilar ones. By contrast, group-based environments exhibit a qualitatively different dynamic. Because exit from a group alters the composition of the group itself, dissatisfaction can propagate transitively, producing cascading exits that rapidly amplify small initial imbalances. The resulting segregation is driven primarily by interaction structure rather than by strong sorting preferences and arises under conditions in which network-based systems would remain relatively mixed. Despite the prevalence of group-based interaction online—forums, subreddits, channels, servers—this mechanism has received comparatively little attention.

This perspective also helps explain why seemingly benign communities can drift toward ideological extremity over time. Interest-based or socially defined spaces—for example, hobby forums or identity-centered communities—may initially attract heterogeneous participants. Yet repeated cycles of selective exit and regrouping can gradually narrow the range of viewpoints expressed, producing ideologically uniform enclaves without deliberate radical intent. The mechanism identified here offers a parsimonious theoretical account of how some online communities become more ideologically homogeneous over time, commonly observed across online subcultures.

Our findings have two implications that challenge common assumptions. First, ideological homogeneity need not be designed, desired, or imposed. It can emerge as an unintended consequence of otherwise ordinary interaction architectures. This reframes echo chambers as emergent system-level phenomena rather than simply as products of individual preferences or platform optimization strategies.

Second, the role of algorithmic filtering is more ambiguous than commonly assumed. In our model, modest personalization can dampen dissatisfaction and slow exit dynamics, thereby stabilizing heterogeneous communities. By contrast, aggressively promoting cross-cutting exposure can increase perceived misalignment and accelerate segregation. This challenges the assumption that more exposure to difference is always beneficial and suggests that certain forms of personalization may, under specific conditions, help sustain diversity rather than undermine it.

At the same time, the model is intentionally stylized, and its aim is not comprehensive explanation. Agent-based models identify plausible mechanisms, not deterministic outcomes. Real platforms involve users with multidimensional identities, overlapping community memberships, and issue-specific engagement that varies in salience and emotional intensity. Platform affordances—ranking systems, moderation practices, interface design—further shape interaction dynamics in ways not captured here. These factors may attenuate, delay, or counteract the segregation dynamics we identify, or give rise to different patterns altogether. Similarly, the empirical analysis is limited to a single subreddit and a single operationalization of semantic distance, and its descriptive design cannot establish the causal mechanisms that the model formalizes.

 

The analysis should therefore be understood as an illustrative plausibility check rather than a test of the model's core dynamics. Importantly, our mechanism is not proposed as a replacement for existing accounts of online homophily, but as a complement: a structural pathway through which segregation can arise even when homophilic preferences and algorithmic filtering are weak or absent.

Despite these limitations, the results carry clear implications for platform governance and public debate. Maintaining ideologically diverse online spaces may be less like engineering a stable equilibrium and more like balancing a dynamic system that is prone to tipping. Interventions that ignore endogenous feedback dynamics risk backfiring, reinforcing the very outcomes they seek to prevent. More broadly, the findings underscore a general principle of digital societies: emergent structural dynamics may play an important and underappreciated role in shaping the political and epistemic consequences of online interaction.

## Author contributions

**Conceptualization:** Petter Törnberg.

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
