## [Decision Letter · Decision Letter 0]

3 Dec 2025

Online Homogeneity Can Emerge Without Filtering Algorithms or Homophily Preferences

PLOS ONE

Dear Dr. Törnberg,

Thank you for submitting your manuscript to PLOS ONE. After careful consideration, we feel that it has merit but does not fully meet PLOS ONE’s publication criteria as it currently stands. Therefore, we invite you to submit a revised version of the manuscript that addresses the points raised during the review process.

Two experts in the field have reviewed the paper. They recognize its potential but indicate that several points require clarification. I agree with their assessment and encourage the author to address their comments carefully. In particular, both reviewers request clearer explanations of some methods used in the study. Reviewer 2 also raises concerns about the paper’s framing, which I recommend the author consider carefully.

We look forward to receiving your revised manuscript.

Kind regards,

Alessandro Galeazzi, PhD

Academic Editor

PLOS ONE

Journal Requirements:

[Dutch Research Council (NWO) VIDI Grant VI.Vidi.231S.089].

Reviewers' comments:

Reviewer's Responses to Questions

**Comments to the Author**

1. Is the manuscript technically sound, and do the data support the conclusions?

Reviewer #1: Yes

Reviewer #2: Yes

2. Has the statistical analysis been performed appropriately and rigorously?

Reviewer #1: Yes

Reviewer #2: No

3. Have the authors made all data underlying the findings in their manuscript fully available?

Reviewer #1: Yes

Reviewer #2: Yes

4. Is the manuscript presented in an intelligible fashion and written in standard English?

Reviewer #1: Yes

Reviewer #2: Yes

Reviewer #1: The paper presents an interesting toy model to represent ideologically aligned communities in social networks. The manuscript is well-written and easy to read. The results are interesting and support the discussions. However, I believe that there are some points that need improvement before publication. In the following, I present my concerns in detail.

1 - On page 4, second line, the author argues that in online social networks, "interaction rarely follows spatial logics". However, people often connect with their offline friends on social media. Consequently, it is not rare to follow spatial logics. For instance, Laniado et al. argue that "friendship ties belonging to denser connected groups tend to arise at shorter spatial distances than social ties established between members belonging to different groups."

See this reference: Laniado, D., Volkovich, Y., Scellato, S., Mascolo, C., & Kaltenbrunner, A. (2018). The impact of geographic distance on online social interactions. Information Systems Frontiers, 20(6), 1203-1218.

2 - The results section starts without describing the parameters used. Therefore, it took me some time to understand the first result. I suggest just adding these parameters to the first paragraph (N=100, C=20, k=10).

3 - Each result is shown for a single (or a few) execution(s) of the dynamics. So, it is not possible to guarantee that the result will always be the same. Performing the experiments more times would make the results more reliable and robust.

4 - Some of the parameters could be better explored (or justified). Why only N=100 and C=20? I suggest exploring these parameters and testing higher values of N.

5 - In the title of Figure 4, there is a variable noise. However, I did not understand its meaning.

6 - I found the first paragraph of Section 3.1 a bit confusing. It says: "We introduce a parameter b controlling the proportion of interaction partnersa bit confusing. It says: "We introduce a parameter b controlling the proportion of interaction partners

who are artificially selected to share the focal agent's opinion."

If I understand correctly, the filter bubble version does not alter the opinions of the agents for subsequent iterations of the model; instead, it modifies the value of "f of interactors". Am I correct? I suggest the author rewrite this paragraph to clarify this information.

Reviewer #2: In this paper the author extends the Schelling model with a simple modification: agents, instead of being located in a geometrical space are located within C abstract communities and can relocate to any other community, without spatial or network constraints. The author shows how this decreases the segregation threshold from 0.3 to 0.1 roughly. Then the author shows how the introduction of a filter bubble effect may actually reduce the segregation (perceived and effective). These results are demonstrated analytically and numerically.

The article is well written and structured, relevant literature was cited, the introduction and the results are clear.

The study seems sound and robust although a methodology section describing the model and the analytical solution and the metrics used would be of help.

The results are interesting and contribute to a timely and ongoing discussion with a well put perspective.

The code is publicly available.

However, there are a number of limitations that need to be addressed

When the author claims that the segregation is not a result of individual choice, what does this mean? There is still a user evaluating their environment and deciding to relocate. Maybe the author could reformulate such claims and just state that there is no intentional homophilic dynamics guiding segregation. Users are not looking for alike individuals, they're just moving away from different ones.

One aspect that I find conceptually unclear is the way the model is set up. We have C communities, and users who can switch communities randomly. While I understand that the goal is to show that segregation can emerge even without intentional behavior, the assumption that users change communities at random is rather strong and somewhat puzzling.

The introduction of the filter-bubble mechanism raises additional questions. As I understand it, users are more likely to see opinions similar to their own within their current community. Yet this seems to reduce segregation, because if users are less exposed to disagreeing opinions, they have fewer incentives to change communities. This could be coherent with some interpretations, but it is not entirely convincing. If the communities are meant to represent real-world communities, for instance, Reddit subreddits, I am not sure such an internal filter-bubble mechanism exists. Once a user is in a subreddit, content is typically shown quite uniformly (subject to popularity), without strong personalization within the community itself. If anything, filter-bubble effects might influence which community a user enters in the first place, not what they see after joining.

Of course, the interpretation would be different if “communities” are intended to represent sets of nodes in a network that are highly connected internally and weakly connected externally. In that case, a network-based model might be more appropriate, or at least this interpretation should be made explicit. Overall, I find it difficult to grasp the intended meaning of the environment and of the filter-bubble mechanism as currently described. Clearer justification, ideally supported by empirical examples, would help substantially.

A related point concerns the description of the resulting dynamics as “surprising” or “counterintuitive.” In fact, the outcome seems quite intuitive: if relocation is driven by exposure to disagreeing opinions, and the algorithm reduces the probability of seeing disagreeing opinions, then it naturally slows down segregation dynamics. This logic has also been discussed in previous work on opinion dynamics on adaptive networks (e.g., Pansanella et al. 2022, Modeling algorithmic bias: simplicial complexes and evolving network topologies, Applied Network Science; for a similar discussion also Valensise, Carlo M., Matteo Cinelli, and Walter Quattrociocchi. "The drivers of online polarization: Fitting models to data." Information Sciences 642 (2023): 119152; maybe also Cinus, Federico, et al. "The effect of people recommenders on echo chambers and polarization." Proceedings of the International AAAI Conference on Web and Social Media. Vol. 16. 2022.).

More generally, I would suggest tempering some of the stronger claims. The model is ultimately a stylized toy model, and we have very limited knowledge of how real platforms actually operate. Making this explicit would, in my view, strengthen the paper. Suggesting that the results explain the segregation found in online environment is an overstatement. First of all the results are not directly compared to any dataset or empirical measure of segregation from different platforms. Second, there are plenty of works in literature that claim to have explained polarization/echo chamber/segregation with their model that are all equally valid and sound. I would suggest some degree of empirical validation to make the claims more robust using dataset from different social media platforms, if possible, otherwise a justification on why it is not possible/does not make sense.

.

Reviewer #1: No

Reviewer #2: **Yes:** Valentina PansanellaValentina PansanellaValentina PansanellaValentina Pansanella

---

## [Author Response · Author response to Decision Letter 1]

9 Dec 2025

Kindly see the attached response letter.

---

## [Decision Letter · Decision Letter 1]

11 Feb 2026

Dear Dr. Törnberg,

Thank you for submitting your manuscript to PLOS ONE. After careful consideration, we feel that it has merit but does not fully meet PLOS ONE’s publication criteria as it currently stands. Therefore, we invite you to submit a revised version of the manuscript that addresses the points raised during the review process.

The reviewers acknowledge the authors’ efforts to improve the manuscript and address previous comments, particularly the addition of an empirical comparison of the results. However, Reviewer 2 has still some concerns regarding the framing of the paper, specifically whether the model design encode mechanism that can be interpreted as human preferences, as well as statements that may be read as implying causal relationships. I agree with the reviewer’s interpretation and recommend that the authors carefully consider these points in order to provide a clearer and more precise framing of their work. In addition, I encourage the authors to thoroughly address the reviewer’s comments regarding the limitations of the model.

We look forward to receiving your revised manuscript.

Kind regards,

Alessandro Galeazzi, PhD

Academic Editor

PLOS One

Journal Requirements:

Reviewers' comments:

Reviewer's Responses to Questions

**Comments to the Author**

Reviewer #1: All comments have been addressed

Reviewer #2: All comments have been addressed

2. Is the manuscript technically sound, and do the data support the conclusions?

Reviewer #1: Yes

Reviewer #2: Yes

3. Has the statistical analysis been performed appropriately and rigorously?

Reviewer #1: Yes

Reviewer #2: Yes

4. Have the authors made all data underlying the findings in their manuscript fully available?

Reviewer #1: Yes

Reviewer #2: Yes

5. Is the manuscript presented in an intelligible fashion and written in standard English?

Reviewer #1: Yes

Reviewer #2: Yes

Reviewer #1: The author has answered all my questions and significantly improved the overall quality of the paper.

Reviewer #2: I thank the author for the careful revision and for engaging seriously with the reviewers’ comments. The manuscript has improved in clarity, and the newly added empirical section is a welcome attempt to connect the theoretical model to observational data. The paper remains well written, clearly structured, and relevant to an active and important debate. The modeling results are sound, well-explained, and supported by both analytical arguments and simulations, and the availability of the code is a strong point.

That said, while many of my first-round concerns have been addressed, some conceptual and empirical issues remain and should be clarified further before publication.

The sentence in the abstract “Taken together, these findings recast online echo chambers not as a result of human preferences or algorithmic design, but as a structural consequence of how online platforms reorganize social interaction,” is conceptually unclear and, as currently phrased, misleading. I interpreted it as “Even in the absence of explicit homophily or personalized recommendation, the basic organizational features of online platforms (bounded communities, easy exit, aggregation of interactions) can generate echo chambers as an emergent outcome”, but this is just an interpretation. However, as I already stated in the first round of review, human preference is still present - even if it is not strong - since exiting due to disagreement and the level of tolerated disagreement are human preferences. In principle, also saying that the dynamic is not a result of “algorithmic design” is not entirely correct: every mechanism related to an online platform is due to algorithmic design, but I think that here one can just change design into personalization, and the meaning becomes clearer. The notion of a “structural consequence” is not defined, and it is not clear what is meant by platforms “reorganizing social interaction”, especially given that the model does not include any active platform intervention. I recommend rephrasing the sentence in a less ambiguous way, stating what the findings suggest, using well-defined terms, and avoiding overstating, attributing agency, or causal mechanisms. The comments on this sentence should be taken as an example to guide revision of the whole text, being careful with terminology and avoiding ambiguous terms that may suggest causality or overstate effects.

The newly added empirical analysis on r/MensRights is a useful addition, but its contribution should be framed more modestly. The logistic regression shows that users whose language is farther from the monthly community centroid are more likely to cease participation in the subsequent time window. This is an interesting and plausible descriptive finding, and it is broadly consistent with the model's micro-level assumption. However, the analysis does not, and cannot, provide evidence for the broader Schelling-type dynamics emphasized in the theoretical sections.

Several limitations deserve more explicit discussion:

- The regression does not control for unobserved user-level heterogeneity (e.g., stable differences in engagement style or propensity to churn), as no user fixed effects or random effects are included. As a result, the estimated association between linguistic distance and exit may reflect selection effects rather than dissatisfaction with minority status.

- Most importantly, linguistic distance from the centroid conflates ideological disagreement with other forms of divergence, such as topic choice or rhetorical style, and is itself endogenously shaped by participation and exit.

- Exit is behaviorally ambiguous and may reflect factors unrelated to dissatisfaction with community composition (e.g., reduced platform use or migration elsewhere).

- The observations are not independent, as users contribute multiple observations and distances are jointly determined through a shared centroid. Because observations are repeated over time for the same users and exit is an absorbing state, the conditional independence assumption of logistic regression is violated.

While the authors do note that the results are “descriptive rather than causal,” some passages nonetheless suggest stronger empirical support for the model’s mechanism than the analysis can justify. I recommend consistently framing this section as an illustrative plausibility check of a single micro-level assumption, rather than as empirical validation of the model’s core dynamics.

Other minor concerns are:

- In the abstract: “as an structural consequence” → as a structural consequence

- “randomly distributed randomly across C communities”: remove redundancies

- Inconsistent use of hyphenation: “micro level”, “macro level” vs. “micro-level”, “macro-level” → standardize to hyphenated forms throughout

- Words such as “sharp,” “striking,” “dramatic,” “paradoxical,” and “counterintuitive” are sometimes overused. While acceptable in moderation, I suggest reducing repetition, especially in the Results and Discussion sections.

- Terminology consistency: “exit,” “disengage,” and “leave” are used somewhat interchangeably. I recommend choosing one primary term (e.g., exit) and using others only when stylistically necessary.

- Avoid causal language. Even small phrases such as “predicts exit” could be replaced, for example, with “is associated with exit” for consistency with the stated descriptive scope.

- I also remain unconvinced by the characterization of the algorithmic filtering dynamics as particularly “counterintuitive” or “paradoxical.” Once the mechanism is explicitly described, the result that reduced exposure to disagreeing opinions slows down exit-driven segregation appears rather intuitive. Moreover, the comparison with Ref. [32] is not entirely accurate. Pansanella et al. show that algorithmic filtering slows down segregation dynamics, whereas in the absence of such filtering echo chamber formation proceeds more rapidly, an outcome that is broadly consistent with the mechanism proposed here. While relocation across communities is not present in that model (nor in Valensise et al.), rewiring links away from disagreeing neighbors represents a closely related mechanism: rather than exiting an entire community, agents exit specific social ties in response to disagreement. In both cases, the removal of disagreeing interactions drives segregation. Although the formation of new links in those models is guided by homophily, the key point remains that algorithmic filtering dampens, rather than accelerates, segregation dynamics, reducing the perceived disagreement and inhibiting the driver of the exit behaviour.

- the z-standardization of the logistic regression could be better clarified, e.g., "All continuous predictors are z-standardized (mean 0, standard deviation 1), so coefficients can be interpreted as the effect of a one–standard-deviation increase in the corresponding variable on the log-odds of exit."

.

Reviewer #1: No

Reviewer #2: **Yes:** Valentina PansanellaValentina PansanellaValentina PansanellaValentina Pansanella

---

## [Author Response · Author response to Decision Letter 2]

10 Mar 2026

Thank you for the opportunity to revise the manuscript. Please see the attached response letter.

---

## [Editor Report · Decision Letter 2]

31 Mar 2026

Echo Chambers Can Emerge Without Algorithmic Personalization or a Preference for Homogeneity

PONE-D-25-52879R2

Dear Dr. Törnberg,

We’re pleased to inform you that your manuscript has been judged scientifically suitable for publication and will be formally accepted for publication once it meets all outstanding technical requirements.

Kind regards,

Alessandro Galeazzi, PhD

Academic Editor

PLOS One

Additional Editor Comments:

I thank the author for thoroughly addressing both the reviewers’ comments and my concerns, significantly improving the clarity of the manuscript. I believe the work is now ready for publication.

---

## [Editor Report · Acceptance letter]

PONE-D-25-52879R2

PLOS One

Dear Dr. Törnberg,

I'm pleased to inform you that your manuscript has been deemed suitable for publication in PLOS One. Congratulations! Your manuscript is now being handed over to our production team.

Kind regards,

on behalf of

Dr. Alessandro Galeazzi

Academic Editor

PLOS One